# WHY SANITY CHECK FOR SALIENCY METRICS FAILS?

## ABSTRACT

Saliency maps are widely leveraged as a post-ad-hoc approach to explain the decision-making process of Deep Learning-based image classification models. However, despite their popularity, ensuring the fidelity of saliency maps remains a complex problem. Researchers have, therefore, introduced saliency metrics to evaluate the fidelity of saliency maps. However, previous studies observed several statistical inconsistencies in the existing saliency metrics without investigating the reason behind the inconsistencies. In this study, we investigate the reason behind the observed statistical inconsistencies. We analyze the inconsistencies by studying the variation in pixel importance ranks, specifically by choosing a case study of varying levels of Gaussian blur (with different $\sigma$ values for the width of the Gaussian Kernel) as the perturbation mechanism. Our findings indicate that the effect of perturbations on prediction probability and pixel importance ranks varies widely across different levels of Gaussian Blur. Consequently, the existing saliency metrics that rely on pixel importance become unreliable for measuring the fidelity of saliency maps. This insight necessitates careful use of saliency metrics and the perturbation technique used while assessing the fidelity of saliency maps in eXplainable AI (XAI). We used Gaussian Blur as our perturbation mechanism, but our approach applies to any perturbation.

## 1 INTRODUCTION

Artificial Intelligence (AI) has been widely adopted across various sectors to solve numerous problems. AI based applications vary in terms of domain, complexity, human and societal impact, among others. Complex models like Deep Learning (DL) and Ensembles normally exhibit higher accuracy as compared to traditional and simple models. However, such achievements have also come at a cost of decreased transparency of the models. Further, in highly regulated fields like health care, insurance, law enforcement, among others there have been concerns regarding the transparency, fairness, privacy and trustworthiness of AI applications due to the black-box nature of deep learning models (Rudin, 2019; Jacovi et al., 2021; Arrieta et al., 2020). This has led to discussions on the widespread adoption of the latest AI models in various sectors (Cubric, 2020; Cam et al., 2019; Güngör, 2020). Considerable research has therefore been dedicated towards explaining the decisions of AI systems under the umbrella of XAI (Arrieta et al., 2020; Selvaraju et al., 2017; Chattopadhay et al., 2018; Zhou et al., 2016; Ramaswamy et al., 2020; Ribeiro et al., 2016; Broniatowski et al., 2021; Lundberg & Lee, 2017).

XAI is often a plethora of techniques across different application domains (Arrieta et al., 2020; Messina et al., 2022). Considering the different mechanisms of XAI, different components can be categorised under four levels as shown in Table 1. Level 1 denotes the model to be explained, Level 2 denotes the XAI system and the saliency maps, Level 3 denotes the saliency metrics to measure the fidelity of saliency maps (Samek et al., 2016; Alvarez Melis & Jaakkola, 2018) and finally Level 4 denotes strategies for measuring the reliability of the saliency metrics from Level 3.

Many works in AI/DL use saliency maps in Level 2 (e.g., Class Activation Maps (CAM)) to explain the decision making process in computer vision tasks (Selvaraju et al., 2017; Chattopadhay et al., 2018).How-

Table 1: Levels in the analysis of saliency maps from XAI systems

| Levels | Description | Examples |
| --- | --- | --- |
| Level 4 | Reliability Checks for Saliency Metrics | Inter-Rater reliability, Inter-Method reliability, Internal Consistency reliability Tomsett et al. (2020) etc |
| Level 3 | Saliency Metrics | Area Over Perturbation Curve Samek et al. (2016), Faithfulness Alvarez Melis & Jaakkola (2018) etc |
| Level 2 | XAI Systems and Saliency or Heatmaps | CAM, Grad-CAM, Guided Grad-CAM, Ablation CAM etc |
| Level 1 | Deep Learning Image Classification Models | Inception_V3, Xception, ResNet50, EfficientNetB7 etc |

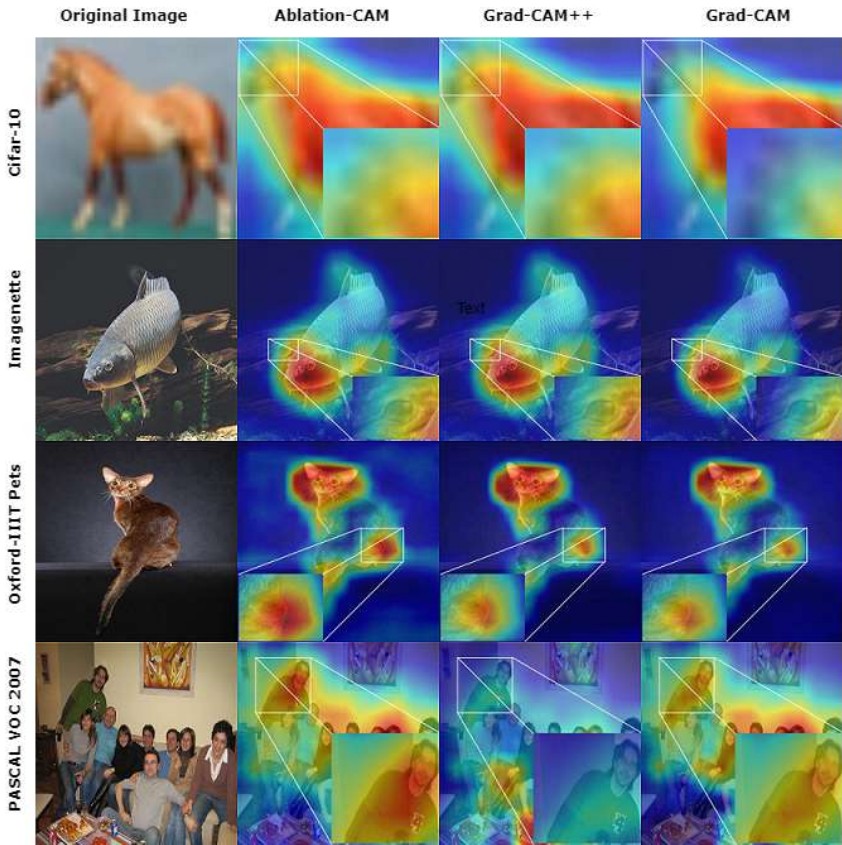

Figure 1: Disagreement between saliency maps generated using Ablation-CAM, Grad-CAM++ and Grad-CAM for ResNet50 model with imagenet weights. Each row represents a randomly chosen image from Cifar-10, Imagenette, Oxford-IIIT Pets and PASCAL VOC 2007 datasets and their corresponding saliency maps

ever, disagreements can be noted among saliency maps generated using different methods for a model and a given image as shown in Figure 1 for AblationCAM (Ramaswamy et al., 2020), GradCAM++ (Chattopadhay et al., 2018) and GradCAM (Selvaraju et al., 2017). Considering these inconsistencies, a set of works

have investigated the fidelity of the generated saliency maps (Selvaraju et al., 2017; Chattopadhay et al., 2018) and have proposed saliency metrics like "Area Over the Perturbation Curve" ($AOPC$) (Samek et al., 2016), Average Drop (AD%), Increase in Confidence (IC%) and Win (W%) (Chattopadhay et al., 2018; Wang et al., 2020) and $faithfulness$ metric Alvarez Melis & Jaakkola (2018) to measure the fidelity of saliency maps (Samek et al., 2016; Bach et al., 2015; Alvarez Melis & Jaakkola, 2018) (Refer Section 2 for details on these metrics).

These saliency metrics were primarily developed as there is no ground-truth to measure against the black-box decisions of DL models. However, such metrics have been criticized because of their unreliability (Tomsett et al., 2020). These metrics rely on perturbations (setting pixels to random values or 0), but do not justify the reason to do so. While such techniques offer implementation simplicity, they may influence image semantics as some pixels are set to 0 or random values changing the image in certain regions. These changes can impact the prediction probability in a non-deterministic manner (Hooker et al., 2019).

Despite having above mentioned saliency metrics which are based on perturbations, these metrics are not studied for various types of perturbations before using them for fidelity analysis. These saliency metrics, rely on the computed importance of pixels (i.e., pixel ranks) for measuring the fidelity of saliency maps. A high variance in pixel ranks for different perturbations would render them unreliable as a fidelity metric. We therefore analyze the impact of perturbations on pixel ranks and study the reliability of the saliency metrics. We argue that the existing saliency metrics are unreliable for measuring the fidelity of saliency maps and are dependent on the type of perturbation used. Although previous work (Tomsett et al., 2020) found unreliability in saliency metrics for validating the fidelity of saliency maps at Level 3, we demonstrate that the reason for such unreliability also lies at the model level (i.e. Level 1). We therefore introduce two new metrics, namely, the Average Ratio of Positive Probability Drops (ARPPD) and the Average Pixel Rank Correlation (APRC) to measure such unreliability at model level (i.e. Level 1). We first present a set of related works in this direction, identity the limitations and list our contributions to address the limitations as presented in next section.

## 2 BACKGROUND AND RELATED WORKS

Saliency Maps is a widely used method to visualize the decision-making process of black box models (Zhou et al., 2016; Selvaraju et al., 2017; Chattopadhay et al., 2018; Ramaswamy et al., 2020; Wang et al., 2020). However, it is difficult to verify the fidelity of such saliency maps as the regions used for making decisions in DL models is not available (annotated) as ground truth. Furthermore, there are inconsistencies amongst saliency maps generated using different methods as shown in Figure 1. The illustration in Figure 1 presents saliency maps on randomly sampled images from the Cifar10, Imagenette, Oxford-IIIT Pets and PASCAL VOC 2007 datasets for pretrained ResNet50 model (imagenet weights) using AblationCAM Ramaswamy et al. (2020), GradCAM++ Chattopadhay et al. (2018) and GradCAM Selvaraju et al. (2017). It can be noted from Figure 1, the saliency maps generated using AblationCAM and GradCAM++ show a high degree of agreement, highlighting the importance of the body, neck, and head of the horse for an image from Cifar-10 dataset (1st row). However, the saliency map generated using GradCAM completely misses highlighting the head of the horse. In the 2nd row it can be observed that AblationCAM and GradCAM++ highlight not only the head of the fish but also other areas in the background as compared to GradCAM. Similarly, the saliency maps generated for the Oxford-IIIT Pets dataset image (3rd row) and PASCAL VOC 2007 image (4th row) show high inconsistency. Thus, the varying level of consistency in the saliency maps, built using various techniques, necessitates the need for having saliency metrics to measure the fidelity of the saliency maps.

A set of works have tried to leverage human annotations for qualitative validation of saliency maps (Selvaraju et al., 2017; Chattopadhay et al., 2018; Wang et al., 2020). These methods at best quantify the saliency map's adherence to expected human perception of the image class and have nothing to do with the fidelity of the saliency map with the model to be explained (Samek et al., 2016; Alvarez Melis & Jaakkola, 2018). As

such we assert that such quantification should not be used to report the fidelity of saliency maps. Due to this limitation of human annotation based approaches, several researchers proposed indirect means of verifying the fidelity of saliency maps (Samek et al., 2016; Alvarez Melis & Jaakkola, 2018; Chattopadhay et al., 2018; Hooker et al., 2019). One such way is to analyze the change in classification probability of an image by perturbing pixels in the sequence of their importance. A metric called "Area Over the Perturbation Curve" ($AOPC$) was proposed by Samek et al. (2016) by extending the work of Bach et al. (2015). The metric is defined below:

$$AOPC_M = \frac{1}{L+1} \left\langle \sum_{k=1}^{L} (f(x_M^{(0)}) - f(x_M^{(k)})) \right\rangle_{p(x)} \qquad (1)$$

where $M$ is the pixel deletion procedure i.e. Most Relevant First (MoRF) or the Least Relevant First (LeRF) procedure.

$f(x)$ is the output value of the classifier for input image $x$ (i.e. the probability assigned to the highest-probability class), $x_M^{(0)}$ is the input image after 0 perturbation steps (i.e. $x_M^{(0)} = x$), $x_M^{(k)}$ is the input image after $k$ perturbation steps, $\langle . \rangle_{p(x)}$ denotes the mean over all images in the data set in a total of $L$ number of pixel deletion steps (Samek et al., 2016).

A set of metrics viz Average Drop (AD%), Increase in Confidence (IC%) and Win (W%) were used by Chattopadhay et al. (2018) and subsequently by Wang et al. (2020) to measure the fidelity of saliency maps. These metrics use the observed change in probability by removing relevant and irrelevant regions alternatively from an image. Another metric, $faithfulness$, was proposed by Alvarez Melis & Jaakkola (2018) which perturbs pixels one by one and notes the change in classification output probability. This differs from the previous metric in which the perturbation is carried out individually for each pixel. As such, the observed change in output probability is the effect of the concerned pixel only. The faithfulness for a single image is calculated as the Pearson correlation between the pixel relevance $R_i$ for pixel $i$ and the change in output probability when $i$ is perturbed. A higher correlation would denote higher faithfulness. Further, the faithfulness for a dataset is calculated by taking the mean of the faithfulness scores of all the images in the dataset.

Another metric, $ROAR$, RemOve And Retrain was proposed by Hooker et al. (2019). The metric is calculated by retraining a model from scratch by perturbing images and then observing the reduction in accuracy. This approach enables the evaluation of fidelity only at a global level and not on individual saliency maps (Tomsett et al., 2020).

## 2.1 LIMITATIONS OF EXISTING WORKS

The main intuition behind the existing metrics, such as AOPC, faithfulness, etc., is that the change in output classification probability is proportional to the relevance of the perturbed pixel. However, this assumes that the pattern of change in output probability (i.e., proportional change in output probability as per the relevance of the perturbed pixel) is similar for different perturbations. If this assumption is not fulfilled, then the metrics' scores would vary for different perturbations, leading to ambiguity. (Tomsett et al., 2020) addressed this issue by noting the difference in results for the perturbation of pixels with 0 vs. a random RGB value. The main problem with perturbations of 0 or a random RGB value is that such perturbations destroy meaningful image semantics. Thus, the change in image semantics would also contribute to the change in classification probability. It is, therefore, challenging to compartmentalize if the change in classification/decision probability results from perturbation or changed image semantics (Hooker et al., 2019).

Some methods leveraged generative models (Agarwal & Nguyen, 2020; Chang et al., 2018) to perform in-painting or remove an object of interest from an image to observe the change in classification probability. This change in classification probability was used to explain the decision of black-box models. These ap-

proaches depend on a generative model to perform in-painting and change the image pixels substantially and cannot be considered perturbations. We, therefore, did not consider such approaches in our analysis as the scope of this paper pertains to perturbations alone.

## 2.2 OUR CONTRIBUTIONS

Our main contributions in this paper are:

- We present a theoretical framework to justify the unreliability of saliency metrics which are based on perturbations. We show that prior to using saliency metrics, the behavior of models w.r.t. to the type of perturbation has to be studied.
- Unlike the previous works which just observed the inconsistencies in saliency metrics, we propose two new metrics to quantify the inconsistencies using prediction probability change and variation of pixel ranks.
- We present an extensive empirical analysis of four widely used DL models and four different datasets to show the applicability of our proposed metrics to measure the inconsistency of the saliency metrics.

## 3 PROPOSED APPROACH

Due to the black box nature of DL models, it is difficult to establish the true importance of pixels (i.e., pixel ranks) used for classifying a particular image. Indirect methods like saliency metrics have been devised to verify the importance of the pixels. However, as discussed in the previous section, these saliency metrics assume that the change in output probability is proportional to the relevance of the perturbed pixel. For this assumption to hold true, the pattern of change (i.e. the proportionate change in output probability as per the relevance of the perturbed pixel) should ideally hold true for all types of perturbations (e.g., Gaussian blurring). In the worst case, the assumption should hold true for a particular perturbation across different hyper parameters of the perturbation mechanism (e.g., various blur factors in Gaussian blurring). The saliency metrics should accompany the perturbation techniques under which they are expected to work reliably. We therefore divide our analysis in three different areas: Section 3.1 presents our theoretical justifications for observed inconsistencies of saliency metrics, the proposed metrics in Section 3.2 quantify the inconsistencies and complement it with a case study of proposed metrics using Gaussian blur as a perturbation mechanism as detailed in Section 3.3.

## 3.1 THEORETICAL FRAMEWORK

Let $R_0$ be the ranks of pixel as per importance obtained from a saliency map. $R_0$ can be expressed as follows:

$$R_0 = \{a_1, a_2, a_3, a_4, \ldots a_i\} \tag{2}$$

where, $R_0$ is the ranked list of pixel importance by any saliency method. $a_1 \rightarrow a_i$ are pixels sorted in the order of their importance i.e. a greater $i$ denotes greater importance.

The assumption on the expected change in output probability by perturbing a pixel can be summarized as:

$$P(I_0) > P(I_{Pert(\sigma, a_i)}) \quad \forall \quad i, \sigma \tag{3}$$

where, $P(.)$ is the predict function of the concerned classification model which takes an image $I$ as input and returns the probability of the top class. $P(I_0)$ is the probability of the top class as predicted for the original i.e. unperturbed image. $I_{Pert(\sigma, i)}$ is an image obtained by perturbing only the $i^{th}$ pixel of an image $I$ with Perturbation mechanism $Pert()$ and hyper-parameter set $\sigma$. We do this by perturbing a copy of the original image and then copying the concerned pixel value from the perturbed image to the original image.

Further, the change in output probabilities of perturbing two relevant pixels $i$ and $j$, where $j$ is more important than $i$, can be summarized given as:

$$\Delta P(I_{Pert(\sigma,a_i)}) < \Delta P(I_{Pert(\sigma,a_j)}) \quad \forall \quad i < j \tag{4}$$

Where, $\Delta P(I_{Pert(\sigma,a_i)}) = P(I_0) - P(I_{Pert(\sigma,a_i)})$

Utilizing Equation (2) and Equation (4) we can generate the list of probability changes, denoted as "$\Delta P(I_{Pert(\sigma)})$", for image $I$ by perturbing each pixel $a_i$ as below:

$$\Delta P(I_{Pert(\sigma)}) = \{(P(I_0) - P(I_{Pert(\sigma,a_1)})), (P(I_0) - P(I_{Pert(\sigma,a_2)})), \cdots$$
$$(P(I_0) - P(I_{Pert(\sigma,a_i)}))\} \tag{5}$$

Where, $P(I_0)$ is the probability of the top class as predicted for the original i.e. unperturbed image.

The probability changes obtained from Equation (5) can be sorted to get an ordered list of pixels. This set of ordered pixels, denoted by $R_\sigma$, represents the importance ranks of the pixels corresponding to $\sigma$. For a perturbation based technique to be applicable in saliency metrics, the pixel importance ranks should ideally be invariant to different sets of hyper-parameters. This invariance to different sets of hyper-parameters is defined as below:

$$\tau(R_{(\sigma=x)}, R_{(\sigma=y)}) = 1 \quad \forall x, y \tag{6}$$

Where, $\tau$ is Kendall's Tau (Kendall, 1945) in our experiments, but it can be any function that calculates the similarity between two rank lists. Further, without the loss of generality we can say that Equation (6) should hold true for any set of pixels obtained from a saliency map. While $R_{(\sigma=x)}$ and $R_{(\sigma=y)}$, in Equation (6), is derived from the same perturbation mechanism $Pert()$ but the same is also applicable when both the pixel rank lists are derived from different perturbation mechanisms.

### 3.2 Proposed Metrics

To validate our argument that the observed change in output probability being proportional to the importance of pixels may not hold true for different perturbations, we pose two critical questions:

1. How does the prediction probability vary after perturbing a pixel with different perturbations and if impacted, to what degree?
2. How does this variation in output prediction probability impact the pixel importance ranks?

In the course of answering these two questions, we propose two new metrics, Average Ratio of Positive Probability Drops (ARPPD) and Average Pixel Rank Correlation (APRC) respectively.

#### 3.2.1 Average Ratio of Positive Probability Drops

The Average Ratio of Positive Probability Drops i.e. $ARPPD_{\mathcal{M}}^{\mathcal{D}_k}$ for an image denotes the compliance with Equation (3) and is defined as:

$$ARPPD_{\mathcal{M}}^{\mathcal{D}_k} = \frac{\sum_{\sigma \in \mathcal{N}} \sum_{j \in \mathcal{S}} [\left(P(I_0^k) - P(I_{Pert(\sigma,a_j)}^k)\right) > 0]}{|\mathcal{N}| \times |\mathcal{S}|} \tag{7}$$

Where, $I^k$ represents the $k^{th}$ image in the dataset $\mathcal{D}$. $\mathcal{M}$ is any DL model to be explained. $\mathcal{N}$ represents the set of sigma values considered in the experiment. $\mathcal{S}$ represents the set of pixels identified for perturbation. Therefore, generalizing this for a complete dataset, Equation (7) can be simplified as Equation (8).

$$ARPPD_{\mathcal{M}}^{\mathcal{D}} = \frac{1}{m} \sum_{k=1}^{m} ARPPD_{\mathcal{M}}^{\mathcal{D}_k} \tag{8}$$

Where $\mathcal{D}$ represents the image dataset, $\mathcal{M}$ is the concerned model and $m$ represents the total number of images in the dataset $\mathcal{D}$.

$ARPPD_{\mathcal{M}}^{\mathcal{D}k}$ and $ARPPD_{\mathcal{M}}^{\mathcal{D}}$ should have an ideal value of 1 but higher values i.e. closer to 1 are better.

### 3.2.2 AVERAGE PIXEL RANK CORRELATION

The metric Average Pixel Rank Correlation $APRC_{\mathcal{M}}^{\mathcal{D}k}$ quantifies the adherence of a model $\mathcal{M}$'s output probability to Equation (6) for the $k^{th}$ image of dataset $\mathcal{D}$ containing $m$ images for different perturbations. Further, $APRC_{\mathcal{M}}^{\mathcal{D}}$ which quantifies the adherence of a model $\mathcal{M}$'s output probability to Equation (6) on a dataset $\mathcal{D}$ with $m$ images for different perturbations. It is calculated by taking the mean of $APRC_{\mathcal{M}}^{\mathcal{D}k}$ for all $m$ images in the dataset $\mathcal{D}$. Formally, $APRC_{\mathcal{M}}^{\mathcal{D}k}$ and $APRC_{\mathcal{M}}^{\mathcal{D}}$ are represented by Equation (9) and Equation (10):

$$APRC_{\mathcal{M}}^{\mathcal{D}k} = \frac{\sum_{x \in \mathcal{N}} \sum_{y \in \mathcal{N}, y \neq x} \tau(R_{(\sigma=x)}, R_{(\sigma=y)})^k}{|\mathcal{N}| \times (|\mathcal{N}| - 1)} \tag{9}$$

$$APRC_{\mathcal{M}}^{\mathcal{D}} = \frac{1}{m} \sum_{k=1}^{m} APRC_{\mathcal{M}}^{\mathcal{D}k} \tag{10}$$

where $m$ represents the number of images in the dataset $\mathcal{D}$, $\tau$ represents Kendall's Tau, $\mathcal{N}$ represents the set of different $\sigma$ values, and $R_{(\sigma=x)}$ and $R_{(\sigma=y)}$ ranked lists built by perturbing an image with different $\sigma$ values for a particular model, $\mathcal{M}$ .

$APRC_{\mathcal{M}}^{\mathcal{D}k}$ and $APRC_{\mathcal{M}}^{\mathcal{D}}$ should have an ideal value of 1 but higher values i.e., closer to 1 are better.

### 3.3 CASE STUDY: GAUSSIAN BLUR

In our analysis we consider Gaussian Blur (with different $\sigma$ as the width of the Gaussian Kernel) as a case study for the mode of perturbation. We use different $\sigma$ values (i.e., $\sigma = 0.2, 0.6$ and 1) to generate perturbed images. Our choice of Gaussian Blur is based on the fact that it preserves the semantics of images compared to other random perturbations with mean or random values of the image. We further choose three random $\sigma$ values guided by two main criteria: the $\sigma$ values should not be very close to each other and they should not be very large. If the $\sigma$ values are too close, the difference in the output probability might be very close, leading to a misleading high APRC score. Further, image semantics will be destroyed if high $\sigma$ values are used, similar to cases of perturbing a pixel with "0" or random value. For our analysis we selected 50 random pixels[1] from each image and use them to calculate the proposed metrics (i.e. ARPPD and APRC). Although our analysis was done using Gaussian Blur, the same setup and approach can be applied to any perturbation mechanism.

## 4 EXPERIMENTAL SETUP

We conducted experiments to study the variance of pixel importance ranks for different levels of Gaussian Blur (with different $\sigma$ values) using four pre-trained image classification models and four well-known datasets. The models we used are InceptionV3 (Szegedy et al., 2016), Xception (Chollet, 2017), ResNet50 (He et al., 2016) and EfficientNet B7 (Tan & Le, 2019) initialized with ImageNet weights. We used the Imagenette from tensorflow.org (et.al.), cifar10 (Krizhevsky, 2009), Oxford-IIIT Pet Dataset (Parkhi et al., 2012) and PASCAL VOC 2007 (Everingham et al.) datasets for our experiments. The Imagenetet dataset is a subset of the Imagenet dataset with ten easily classified classes. We have used the validation part of this

---

[1]The justification for randomly selecting pixels for the analysis is provided in Appendix A0.1.

dataset for our experiments, which has around 3925 images. Similarly, for the cifar10 dataset, we used the validation part with 10000 images. The Oxford Pet and PASCAL VOC 2007 datasets did not have train and test splits. Hence, we considered all the images for these two datasets, i.e., 7390 of the Oxford Pet dataset and 4952 of the PASCAL VOC 2007 dataset. In total, we used 26267 images for our analysis. For each model, $predict$ was called for $26267\ images \times 50\ pixels \times\ 3\ \sigma$ values, i.e., more than 3.9 million times. Further, our goal was not to be exhaustive with different datasets and models but to understand the impact of perturbations to evaluate the fidelity of saliency maps from the perspective of pixel ranking. So, this number of images was deemed sufficient for our analysis. Our code [2] was written in Python 3.10 and Tensorflow 2.9 and for computing we leveraged P100 GPUs.

## 5 RESULTS AND DISCUSSION

Table 2: Average ARPPD and APRC values across all images of Imagenette, Cifar-10, Oxford-IIIT Pet and PASCAL VOC2007 datasets with various models.The results are shown as Mean (Standard Deviation). Ideal value ARPPD and APRC should be 1 and higher the better.

| | ARPPD | | | | APRC | | | |
|---|---|---|---|---|---|---|---|---|
| Dataset | I_V3 | Xc | R_50 | E_B7 | I_V3 | Xc | R_50 | E_B7 |
| Imagenette | 0.63 (0.22) | 0.62 (0.21) | 0.59 (0.17) | 0.64 (0.20) | 0.35 (0.43) | 0.33 (0.41) | 0.26 (0.39) | 0.28 (0.39) |
| Cifar-10 | 0.69 (0.16) | 0.69 (0.16) | 0.68 (0.20) | 0.83 (0.12) | 0.24 (0.35) | 0.17 (0.29) | 0.20 (0.30) | 0.19 (0.32) |
| Oxford Pets | 0.56 (0.19) | 0.58 (0.21) | 0.58 (0.17) | 0.60 (0.20) | 0.28 (0.39) | 0.29 (0.40) | 0.27 (0.38) | 0.28 (0.39) |
| VOC2007 | 0.53 (0.18) | 0.55 (0.18) | 0.58 (0.17) | 0.57 (0.19) | 0.27 (0.38) | 0.28 (0.38) | 0.26 (0.38) | 0.27 (0.39) |

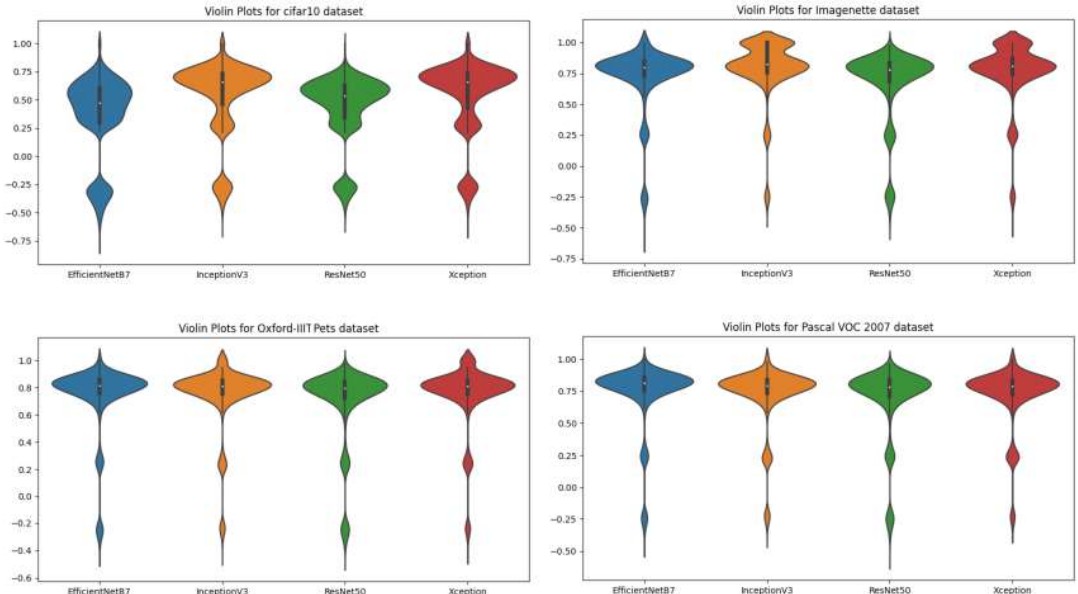

Figure 2: Violin Plots of APRC values for all the models across all the datasets

Table 2 shows the ARPPD values for different models over different datasets. The chosen models i.e. Inception_V3, Xception, ResNet50 and EfficientNetB7 are represented by I_V3, Xc, R_50 and E_B7 respectively.

[2]Available at https://github.com/perturbations/saliencymetrics_unreliability.git

As seen from Table 2, it can be observed that the ARPPD values are around 0.6 for Inception_V3, Xception and ResNet50 while the same for EfficientNetB7 is around 0.8. This indicates that EfficientNetB7 is much more compliant with Equation (4) on the mentioned datasets as compared to Inception_V3, Xception and ResNet50. However, this needs further investigations.

Table 2 shows the APRC values for all the models over all the datasets. As seen from the table, the APRC values are small but as per Equation (4), they should have been equal to 1. This is the same for all four models and across all the datasets [3]. Figure 2 shows the violin plots of the APRC values for all the models across all the datasets. As seen in the plot all the models across all the datasets have a large number of instances where the APRC values are much lower than 1. The same can be observed in the pairwise APRC values for the rank lists obtained for each perturbation type (i.e. different sigmas)[4].

Figure 2 shows the violin plots of the APRC values for all the models across all the datasets. As seen in the plot, all the models across all the datasets have a large number of instances where the APRC values are much lower than 1.

As seen from Table 2, APRC values are low at each class level and also over the whole dataset [5]. This indicates that there is high variance in the ranks of the pixels for the same model and the same image for different types of perturbations. This would mean that the same pixel would be given different ranks for different perturbations. Thus, the idea of verifying pixel importance ranks of saliency maps leveraging perturbations would be unreliable due to this high variance in pixel ranks. Consequently, all metrics like $AOPC$ and $faithfulness$ etc. that implicitly rely on the invariance of pixel ranks would be rendered ineffective in verifying the fidelity of saliency maps.

## 5.1 LIMITATIONS OF OUR WORK

While it is beyond the scope of this study but the reason for the observed variance in the ranks could be the result of the lack of robustness around the predicted instance of the concerned DL models. Further experiments will have to be conducted to validate the same.

## 6 CONCLUSION AND FUTURE WORK

The prediction probability of DL models varied for a given pixel of the same image, and the same model differs significantly due to perturbations (i.e., Gaussian Blur with different $\sigma$). This variation in the output probabilities also leads to a high variance in the calculated pixel importance ranks. Thus, the metrics that implicitly rely on the invariance of pixel importance rank for measuring fidelity would be rendered unreliable and fail the sanity checks. While previous studies have limited the analysis of unreliability to the metric level, we have demonstrated that unreliability arises as a property of the DL models with respect to perturbations. Furthermore, the proposed APRC score, averaged over multiple runs, can be leveraged to measure the readiness of a model (for a particular perturbation with different hyper-parameters or different perturbation types) for fidelity analysis. Thus, we recommend leveraging the APRC score averaged over multiple runs as a preconditional check before analyzing the fidelity of saliency maps. Future works should consider the high variance in pixel ranks and lack of robustness around the predicted instance for devising reliable saliency metrics. In the future, we plan to study the behavior of adversarially trained DL models for perturbations using the proposed metrics.

---

[3]Table A1, Table A2 and Table A3 in the Appendix shows the APRC values for the various classes of the Imagenette, Cifar-10 and Oxford-IIIT Pets datasets.

[4]Please refer to Figure A1, Figure A2, Figure A3 and Figure A4 provided in Appendix.

[5]Class wise results can be obtained from Table A1, Table A2 and Table A3 in Appendix.

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
