# A APPENDIX: WHY SANITY CHECK FOR SALIENCY METRICS FAILS?

### A0.1 PIXEL SELECTION AND RANKING

Selection of pixels for our analysis is another critical aspect for our analysis. As the size of the images are typically $299 \times 299$, $224 \times 224$ or $600 \times 600$ pixels, it is computationally expensive to conduct an analysis on all pixels. We therefore conduct our analysis on a subset of pixels which were randomly selected. Our approach to randomly select the pixels can be further justified from a theoretical perspective as explained below.

Let $Q$ be a set of pixels such that $|Q| > 1$. We can define a hypothetical function $\psi(Q)$ that measures the importance of $Q$ for the decision making process of the model as:

$$\psi : Q \rightarrow \{1, 2, \ldots, |Q|\} \subseteq \mathbb{R}$$

where $\mathbb{R}$ is the set of all real numbers and a greater value of $\psi(Q)$ indicates greater importance.

Given Equation (2), we can define an image $\mathbb{A}$ as an ordered set of pixels sorted according to their importance using function $\psi$.

$$\mathbb{A} = \{a_1^u, a_2^v, a_3^w, \ldots a_i^z\} \tag{11}$$

where, $R_0$ is the ordered set of pixels. $1 \rightarrow i$ are importance for the pixel index/ids $u \rightarrow z$ generates by $\psi$ i.e. $\psi(a^u) = 1$, $\psi(a^v) = 2 \ldots \psi(a^z) = i$ etc, where a greater value of $\psi(Q)$ indicates greater importance of the pixel set $Q$ in the image.

Let us assume that $\mathbb{B}$ is a randomly selected subset of pixels. Thus $\mathbb{B}$ can be defined as below:

$$\mathbb{B} = \{a_1^x, a_2^y, a_3^z, \ldots a_j^n\} \subseteq \mathbb{A} \quad \text{s.t.}$$
$$a^e \neq a^f \quad \text{for} \quad e \neq f \tag{12}$$

where $e$ and $f$ are two random pixels. Let us assume that the order of pixels in $\mathbb{A}$ and $\mathbb{B}$ are different. This implies according to induction:

$$\exists \quad (a^p, a^q) \in \mathbb{B} \quad \text{s.t.}$$
$$\psi(a^p) > \psi(a^q) \in \mathbb{B} \quad \wedge \quad \psi(a^p) < \psi(a^q) \in \mathbb{A} \tag{13}$$

However, $\psi(a^p) > \psi(a^q) \in \mathbb{B}$ and $\psi(a^p) < \psi(a^q) \in \mathbb{A}$ cannot be true at the same time, we can by mathematical induction deduce that $\nexists \quad (a^p, a^q) \in \mathbb{B}$ that satisfy both conditions given in Equation (13). As such the order of pixels as per their importance are same in both $\mathbb{A}$ and $\mathbb{B}$. We leverage this property that the order of importance of the pixels do not change even in randomly selected (without repetition) subsets for our analysis. If the selected pixels have the same importance ranks, their relative orders are not considered to affect the rank correlation.

Further, it has to be noted that the $APRC$ metric can also be used to measure the readiness of a model (for a particular perturbation with different hyper-parameters or different perturbation types) for analysis of its saliency maps (created using any XAI system) if it is averaged for multiple rounds. A model with higher $APRC$ score over multiple runs would enable more reliable verification of its saliency maps as compared to a model with lower $APRC$ score. However, since the goal of our paper was not to benchmark models based on their readiness score hence we did not venture in this direction.

### A0.2   CLASSWISE APRC SCORES FOR IMAGENETTE, CIFAR-10 AND OXFORD-IIIT PET DATASET

| Dataset | I_V3 | Xc | R_50 | E_B7 |
|---|---|---|---|---|
| n01440764 | 0.357 (0.427) | 0.31 (0.406) | 0.273 (0.385) | 0.288 (0.396) |
| n02102040 | 0.285 (0.392) | 0.314(0.405) | 0.268 (0.386) | 0.277 (0.392) |
| n02979186 | 0.261 (0.368) | 0.261(0.365) | 0.255 (0.371) | 0.258 (0.366) |
| n03000684 | 0.487 (0.459) | 0.433(0.45 ) | 0.264 (0.379) | 0.27 (0.382) |
| n03028079 | 0.286 (0.389) | 0.287(0.39 ) | 0.267 (0.38 ) | 0.275 (0.383) |
| n03394916 | 0.341 (0.423) | 0.309(0.404) | 0.269 (0.381) | 0.27 (0.382) |
| n03417042 | 0.303 (0.398) | 0.358(0.423) | 0.266 (0.378) | 0.278 (0.382) |
| n03425413 | 0.371 (0.431) | 0.341(0.413) | 0.26 (0.372) | 0.269 (0.382) |
| n03445777 | 0.515 (0.473) | 0.365(0.434) | 0.259 (0.38 ) | 0.288 (0.397) |
| n03888257 | 0.425 (0.457) | 0.318(0.411) | 0.248 (0.364) | 0.294 (0.403) |

Table A1: APRC scores in the format Mean(Standard Deviation) across various classes of Imagenette Dataset with various models.

| Dataset | I_V3 | Xc | R_50 | E_B7 |
|---|---|---|---|---|
| airplane | 0.233 (0.342) | 0.229 (0.34 ) | 0.179 (0.28 ) | 0.187 (0.318) |
| automobile | 0.232 (0.337) | 0.233 (0.339) | 0.203 (0.299) | 0.186 (0.314) |
| bird | 0.229 (0.341) | 0.233 (0.341) | 0.188 (0.292) | 0.181 (0.313) |
| cat | 0.230 (0.341) | 0.238 (0.341) | 0.194 (0.294) | 0.187 (0.316) |
| deer | 0.236 (0.344) | 0.241 (0.344) | 0.201 (0.295) | 0.186 (0.309) |
| dog | 0.229 (0.343) | 0.239 (0.346) | 0.198 (0.296) | 0.188 (0.313) |
| frog | 0.243 (0.348) | 0.24 (0.349) | 0.204 (0.302) | 0.186 (0.318) |
| horse | 0.249 (0.356) | 0.259 (0.358) | 0.195 (0.304) | 0.178 (0.315) |
| ship | 0.24 (0.346) | 0.242 (0.345) | 0.186 (0.285) | 0.192 (0.315) |
| truck | 0.242 (0.349) | 0.24 (0.345) | 0.198 (0.302) | 0.188 (0.313) |

Table A2: APRC scores in the format Mean(Standard Deviation) across various classes of Cifar-10 Dataset with various models.

| Dataset | I_V3 | Xc | R_50 | E_B7 |
|---|---|---|---|---|
| English Cocker Spaniel | 0.271(0.383) | 0.274 (0.379) | 0.268 (0.386) | 0.273 (0.389) |
| Keeshond | 0.271(0.388) | 0.273 (0.385) | 0.273 (0.39 ) | 0.274 (0.39 ) |
| Samoyed | 0.272(0.386) | 0.272 (0.383) | 0.272 (0.386) | 0.277 (0.391) |
| German Shorthaired | 0.264(0.384) | 0.273 (0.375) | 0.267 (0.39 ) | 0.267 (0.389) |
| Havanese | 0.272(0.376) | 0.275 (0.38 ) | 0.271 (0.39 ) | 0.277 (0.391) |
| American Pit Bull Terrier | 0.268(0.376) | 0.273 (0.378) | 0.262 (0.386) | 0.267 (0.39 ) |
| Maine Coon | 0.271(0.386) | 0.277 (0.382) | 0.27 (0.384) | 0.27 (0.392) |
| Scottish Terrier | 0.276(0.381) | 0.274 (0.38 ) | 0.267 (0.39 ) | 0.274 (0.388) |
| Bengal | 0.271(0.383) | 0.27 (0.379) | 0.266 (0.378) | 0.274 (0.388) |
| Russian Blue | 0.272(0.386) | 0.272 (0.387) | 0.267 (0.375) | 0.269 (0.398) |
| Shiba Inu | 0.277(0.381) | 0.272 (0.384) | 0.264 (0.393) | 0.283 (0.394) |
| Basset Hound | 0.268(0.384) | 0.274 (0.381) | 0.265 (0.385) | 0.273 (0.386) |
| Chihuahua | 0.272(0.383) | 0.268 (0.385) | 0.263 (0.379) | 0.272 (0.376) |
| Saint Bernard | 0.27(0.382 ) | 0.267 (0.385) | 0.27 (0.387) | 0.276 (0.389) |
| Persian | 0.27(0.387 ) | 0.273 (0.386) | 0.267 (0.386) | 0.268 (0.388) |
| Abyssinian | 0.268(0.381) | 0.272 (0.379) | 0.263 (0.374) | 0.264 (0.384) |
| Boxer | 0.276(0.38 ) | 0.269 (0.383) | 0.266 (0.384) | 0.275 (0.393) |
| Great Pyrenees | 0.274(0.384) | 0.274 (0.387) | 0.266 (0.385) | 0.277 (0.392) |
| Egyptian Mau | 0.266(0.385) | 0.271 (0.381) | 0.269 (0.373) | 0.261 (0.384) |
| Leonberger | 0.274(0.384) | 0.272 (0.385) | 0.27 (0.386) | 0.277 (0.392) |
| Miniature Pinscher | 0.273(0.382) | 0.265 (0.385) | 0.261 (0.385) | 0.264 (0.386) |
| Beagle | 0.271(0.383) | 0.268 (0.379) | 0.271 (0.378) | 0.269 (0.386) |
| Ragdoll | 0.276(0.387) | 0.27 (0.386) | 0.262 (0.38 ) | 0.281 (0.394) |
| Bombay | 0.268(0.38 ) | 0.269 (0.376) | 0.264 (0.371) | 0.27 (0.386) |
| Yorkshire Terrier | 0.269(0.38 ) | 0.276 (0.379) | 0.269 (0.392) | 0.272 (0.385) |
| English Setter | 0.272(0.382) | 0.269 (0.38 ) | 0.271 (0.385) | 0.277 (0.392) |
| Pomeranian | 0.27 (0.39 ) | 0.271(0.386) | 0.269 (0.377) | 0.271 (0.383) |
| Japanese Chin | 0.269(0.385) | 0.27 (0.386) | 0.265 (0.389) | 0.277 (0.388) |
| American Bulldog | 0.275(0.378) | 0.27 (0.382) | 0.269 (0.383) | 0.269 (0.398) |
| Pug | 0.267(0.383) | 0.266 (0.384) | 0.267 (0.38 ) | 0.267 (0.381) |
| British Shorthair | 0.27 (0.385) | 0.274(0.385) | 0.266 (0.378) | 0.28 (0.39 ) |
| Siamese | 0.274(0.387) | 0.274 (0.385) | 0.264 (0.379) | 0.266 (0.386) |
| Wheaten Terrier | 0.275(0.386) | 0.28 (0.381) | 0.267 (0.388) | 0.273 (0.391) |
| Sphynx | 0.27 (0.38 ) | 0.273(0.382) | 0.258 (0.373) | 0.271 (0.389) |
| Staffordshire Bull Terrier | 0.27 (0.385) | 0.274(0.376) | 0.269 (0.385) | 0.277 (0.392) |
| Birman | 0.274(0.387) | 0.27 (0.384) | 0.263 (0.379) | 0.273 (0.391) |
| Newfoundland | 0.278(0.384) | 0.273 (0.383) | 0.28 (0.385) | 0.278 (0.39 ) |

Table A3: APRC scores in the format Mean(Standard Deviation) across various classes of Oxford-IIIT Pet Dataset with various models.

### A0.3 Violin plots of Pairwise APRC scores

The violin plots in Figure A1, Figure A2, Figure A3 and Figure A4 represent the various pairwise APRC values. For example the violin plot corresponding to $E\_B7$ $0.2$ $v/s$ $0.6$ represents the distribution of the APRC values calculated for the rank lists obtained by perturbation of $\sigma = 0.2$ and $\sigma = 0.6$ for the EfficientNet B7 model on the respective datasets. As seen from the pairwise violin plots it is seen that the APRC values are much lower than 1. We notice that the pairwise APRC values for the rank lists corresponding to $\sigma = 0.6$ and $\sigma = 1$ are closer to 1 than the other two pairwise values. However, such differences in pairwise APRC values confirm our previous results that there is a high variance in pixel ranks for different perturbation types.

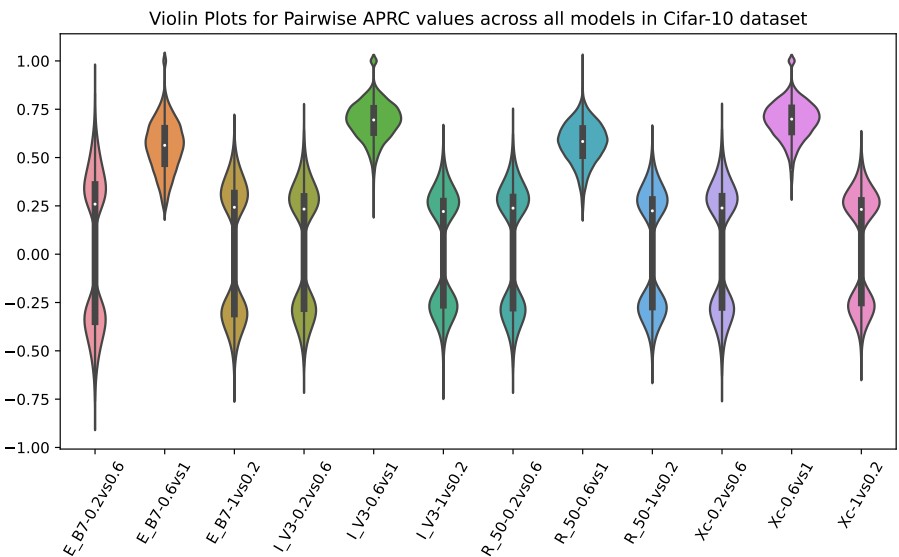

Figure A1: Violin Plots of pairwise APRC values for all ranked lists corresponding to each perturbation(i.e. sigma) of all models for cifar10 dataset

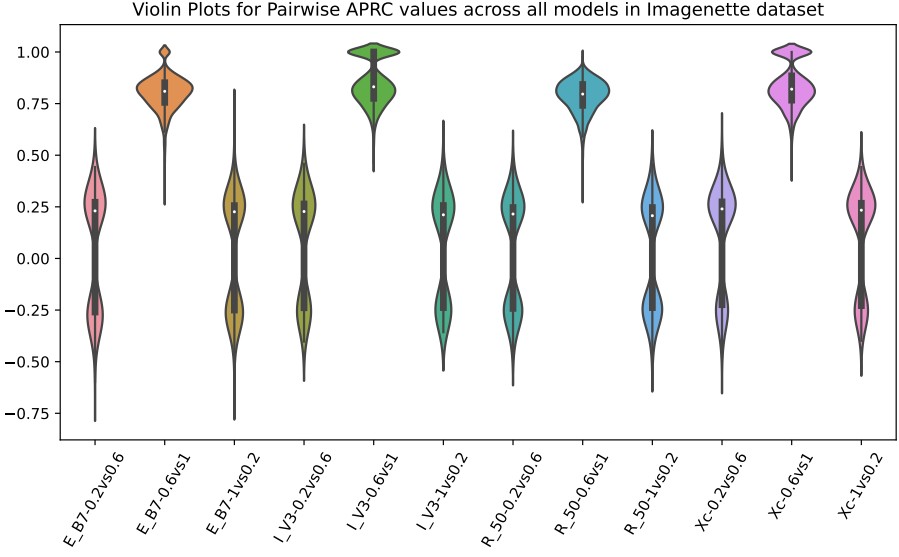

Figure A2: Violin Plots of pairwise APRC values for all ranked lists corresponding to each perturbation(i.e. $\sigma$) of all models for Imagenette dataset

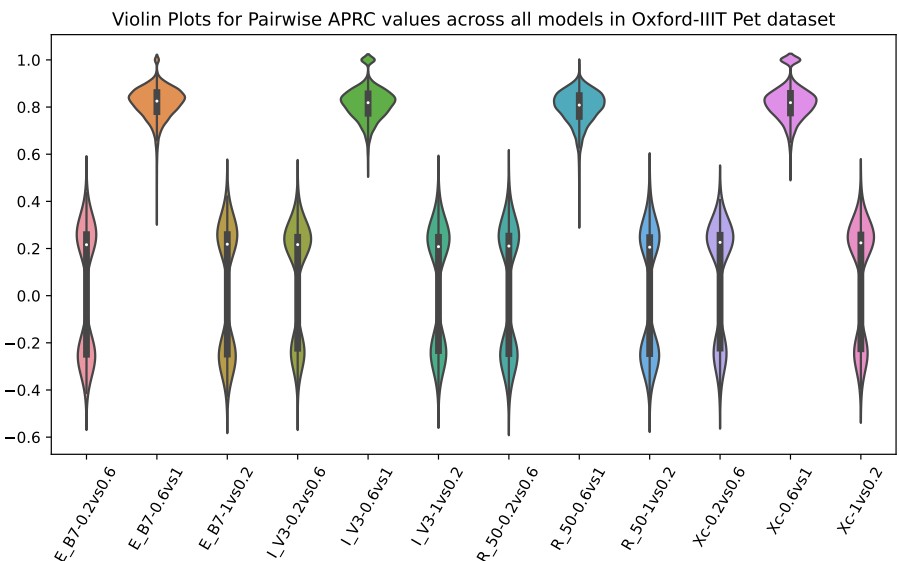

Figure A3: Violin Plots of pairwise APRC values for all ranked lists corresponding to each perturbation(i.e. $\sigma$) of all models for Oxford-IIIT Pet dataset

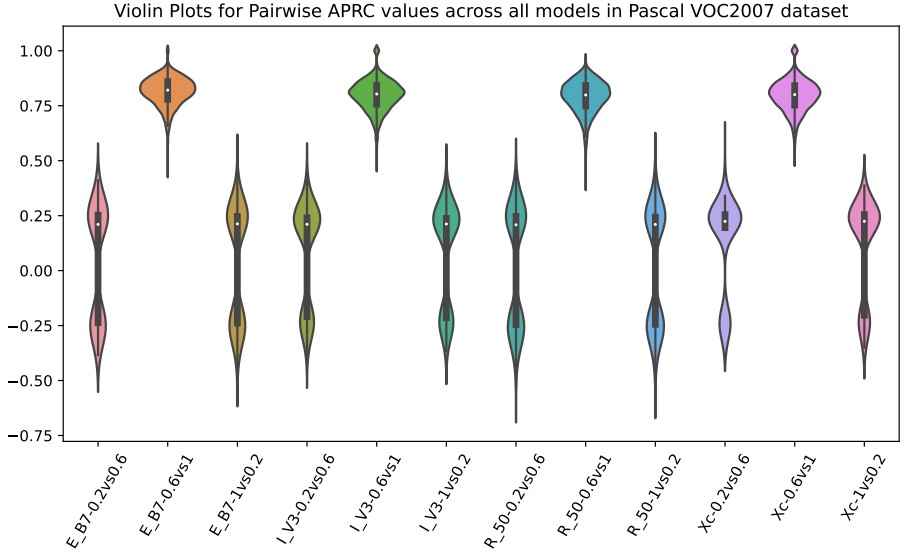

Figure A4: Violin Plots of pairwise APRC values for all ranked lists corresponding to each perturbation(i.e. $\sigma$) of all models for PASCAL VOC2007 dataset