# OpenReview forum: "Why Sanity Check for Saliency Metrics Fails?"
_ICLR.cc/2024/Conference — Submitted to ICLR 2024_

### Official Review · Reviewer_VUjF · 2023-10-30

**Soundness:** 2 fair
**Presentation:** 1 poor
**Contribution:** 1 poor
**Rating:** 1
**Confidence:** 4

**Summary:**

This paper investigates perturbation-based metrics for evaluating the reliability of saliency maps for  image classification explanations. Perturbation-based approaches “delete” pixels to evaluate how much each pixel contributes to the model decision, but the method used to “delete” pixels (e.g., replacing them with 0s vs. random noise) has an impact on the evaluation result. This paper demonstrates that impact using blur as the deletion method and proposes a metric to measure how much the choice of replacement value in a perturbation analysis will affect the result.

**Strengths:**

There are some interesting ideas in this paper -- it does seem worthwhile to do a more in-depth analysis of how the choice of replacement value affects the results in perturbation-based methods.

**Weaknesses:**

The writing is frequently unclear, which makes many sections of the paper difficult to understand.

The explanation of the problem and previous literature is fairly shallow and sometimes incorrect. For example, pg 3 para 2 claims: “These metrics rely on perturbations (setting pixels to random values or 0), but do not justify the reason to do so.” The justification is that these perturbations “delete” the information in those pixels by replacing them with values that have no relation to the image class.

The main finding that different pixel replacements will produce different results in perturbation-based methods is unsurprising and was previously demonstrated in Tomsett et al., 2020.

I’m not sure the proposed metric has wide application, and this study isn’t sufficient to validate it – it should be validated on a wider range of perturbations (including methods like inpainting or replacing pixels with values drawn from the same/different classes).

**Questions:**

“Our choice of Gaussian Blur is based on the fact that it preserves the semantics of images compared to other random perturbations with mean or random values of the image.” This isn’t true for the CNN models used in this paper – blurring images significantly reduces the model’s ability to recognize the image, similar to adding pixel noise (Geirhos et al., 2018: https://arxiv.org/pdf/1808.08750.pdf).

Is the “Imagenette” dataset used in this paper a typo for ImageNet?

Why exclude inpainting / generative models from the set of possible perturbations? This seems like a completely valid type of perturbation that is very relevant if the goal is to show that different perturbations will have different

---

> ### Author Response · Authors · 2023-11-20
> **Rebuttal: Official Review of Submission4212 by Reviewer VUjF**
>
> Overall Comment: We thank the reviewer for the helpful comments and suggestions.
>
> W1: "The writing is frequently unclear, which makes many sections of the paper difficult to understand."
>
> Author's comment: Thanks for the suggestion. We will work on the language and make it more readable in the final version.
>
> W2: "The explanation of the problem and previous literature is fairly shallow and sometimes incorrect. For example, pg 3 para 2 claims: “These metrics rely on perturbations (setting pixels to random values or 0), but do not justify the reason to do so.” The justification is that these perturbations “delete” the information in those pixels by replacing them with values that have no relation to the image class."
>
> Author's comment: We discuss it in Section 2.1, "The main problem with perturbations of 0 or a random RGB value is that such perturbations destroy meaningful image semantics. Thus, the change in image semantics would also contribute to the change in classification probability. It is, therefore, challenging to compartmentalize if the change in classification/decision
> probability results from perturbation or changed image semantics (Hooker et al., 2019).". Our work is motivated by the observations of Hooker et al., 2019 who, in their study, proved that replacing image pixels with random value or "0" will have unwanted impact on the classification probability.
>
> W3: "The main finding that different pixel replacements will produce different results in perturbation-based methods is unsurprising and was previously demonstrated in Tomsett et al., 2020."
>
> Author's comment: We believe the reviewer has misinterpreted our contribution. We do not dispute the obvservation so Tomsett et al., 2020 as discussed in page no 3, paragraph no 3. We try to investigate why such inconsistencies were observed by Tomsett et al., 2020 (refer Section 2.1 and Section 3) as our main contribution.
>
> W4: "I’m not sure the proposed metric has wide application, and this study isn’t sufficient to validate it – it should be validated on a wider range of perturbations (including methods like inpainting or replacing pixels with values drawn from the same/different classes)."
>
> Author's comment: Our goal, in this paper, was to investigate the reason behind the inconsistencies observed by Tomsett et al., 2020. For this, we proposed the metrics ARPPD and APRC. Any fidelity analysis without investigating ARPPD and APRC would be ambiguous (as discussed in Section 3 and experimentally supported in Section 5). We show the applicability of the metrics by a case study of Gaussian Blur. However, our framework can be easily extended to study any other perturbation for fidelity analysis.
>
> Q1: "“Our choice of Gaussian Blur is based on the fact that it preserves the semantics of images compared to other random perturbations with mean or random values of the image.” This isn’t true for the CNN models used in this paper – blurring images significantly reduces the model’s ability to recognize the image, similar to adding pixel noise (Geirhos et al., 2018: https://arxiv.org/pdf/1808.08750.pdf)."
>
> Author's comment: Please refer to Author's comment for W2. The provided reference talks about the drop in classification accuracy for different values of hyper parameters for different types of perturbations. As illustrated in Figure 3(c) and 3(d) in Geirhos et al., 2018, the drop in classification accuracy for blur decays much slower than that of uniform noise indicating that blurring images does not significantly reduce the model's ability to recognize the image for lower sigma values.
> Further, it is not enough to draw the conclusion that blur has similar effect on the drop in output probability as compared to other perturbation methods like mean or random values, both of which have significant effect of introducing perturbations that can change the meaning of image. Further, in scenarios where the output probability for the top class is close to the next class, a small change in the output probability might flip the class and increase the misclassification rate without significant change in the output probability.
>
> Q2: "Is the “Imagenette” dataset used in this paper a typo for ImageNet?"
> Author's Comment: As stated in the Section 4 Experimental Setup, Imagenette is a subset of the ImageNet Dataset with 10 easily identifiable classes. Hence, it is not a typo.
>
> Q3: "Why exclude inpainting / generative models from the set of possible perturbations? This seems like a completely valid type of perturbation that is very relevant if the goal is to show that different perturbations will have different"
>
> Author's comments: Please refer to Author's comment for W4.

---

> > ### Comment · Reviewer_VUjF · 2023-11-22
> > **Response to authors**
> >
> > Thanks for the response.
> >
> > To respond to your clarification about the main goal/contribution of the paper, namely:
> > "We try to investigate why such inconsistencies were observed by Tomsett et al., 2020 (refer Section 2.1 and Section 3) as our main contribution."
> >
> > I don't agree that this is a good thing to investigate -- I think it's already very well understood why different perturbations have different effects. It's just hard to predict precisely what effect a given perturbation will have.
> >
> > Either way, if this is the goal/contribution, then it would be important to investigate a much wider range of perturbations than just blur, including generative model perturbations.

---

> > > ### Author Response · Authors · 2023-11-22
> > > **Response to reviewer's comments**
> > >
> > > We thank the reviewer for reviewing our comments and putting the next set of comments. We humbly put our comments as below:
> > >
> > > Reviewer's comment: "I don't agree that this is a good thing to investigate -- I think it's already very well understood why different perturbations have different effects. It's just hard to predict precisely what effect a given perturbation will have."
> > >
> > > Author's comment: As per our knowledge there is no work which formulates a way to understand the impact of perturbations on the inconsistency of saliency metrics. The nearest work, Tomsett et. al. 2020 reports the observation of inconsistency in saliency metrics but does not explain the reasons behind it. However, since the reviewer is aware of literatures which study and explain the effect of perturbations on the inconsistency of saliency metrics, we humbly request the reviewer to provide them.
> > >
> > > Reviewer's comment: "Either way, if this is the goal/contribution, then it would be important to investigate a much wider range of perturbations than just blur, including generative model perturbations."
> > >
> > > Author's comment: We agree with the reviewer that it would be interesting to see the impact of various types of perturbations on different models and datasets. In the near future we intend to do a survery for it. However, we would like to emphasize that even for any other perturbation type, our proposed metrics ARPPD and APRC will be used to explain the inconsistency of saliency metrics. One of the strengths of our approach (i.e. using the proposed metrics) is that we can bypass analyzing the saliency maps from different CAMs and directly use the proposed metrics to evaluate the inconsistency of the saliency metrics. Further, we dont see any scenario in which low values of ARPPD and APRC could indicate better consistency of saliency metrics or vice versa. However, since the review thinks otherwise, hence, we humbly request the reviewer to point out such scenarios.

---

> > > > ### Comment · Reviewer_VUjF · 2023-11-23
> > > > **Response to authors**
> > > >
> > > > Different perturbations produce different results because they change the image in different ways, and thus are interpretted in different ways by the neural network.
> > > >
> > > > For example, suppose instead of replacing pixels with 0 (black) you replaced them with specific colours (e.g., red, or blue, or green). You should expect each of these colours to have a different result, because they will change the model's decision-making process in different ways depending on the image/perturbed area and the ground truth class/potential alternate classes. If you perturb an image of an apple by replacing the apple pixels with black or blue pixels that should change the model's decision a lot. If you replace the apple with red or yellow pixels, that might not change the model's decision much at all, since a blob or red or yellow pixels might still be interpretted as an apple.
> > > >
> > > > I don't know if there's any publication that specifically states this, since it follows straightforwardly from how neural networks work (or how any vision method works). It's not the kind of thing you would explain in a paper. It's a bit like asking for a citation for the claim that 25 is greater than 3 -- I don't think you can find a publication that specifically states that, it just follows straightforwardly from basic math.
> > > >
> > > > As I said before, predicting exactly *how* a specific perturbation will change the model's decision is not well understood. But *why* it changes is well understood.

---

> ### Author Response · Authors · 2023-11-23
> **Response to Reviewer's comment**
>
> We thank the reviewer for going through our comments and replying with additional comments. Please find our response below:
>
> We agree with the reviewer's statement:
>
> > "predicting exactly how a specific perturbation will change the model's decision is not well understood."   ---- (1)
>
> > "But why it changes is well understood."  -------------(2)
>
> Infact point(1) is the main basis of our study.
>
> As stated in section 2.1, perturbation-based saliency metrics make an implicit assumption regarding how perturbations change the model's decision (i.e., the drop in output probability is proportional to the relevance/importance of the pixel).  \----(3)
>
> The assumption in point (3) has two main parts viz:
>
> 1. There will always be a positive drop (i.e., Prob(unperturbed_image) - Prob(perturbed_image) > 0) in output probability if a relevant pixel is removed (i.e. perturbed as removing or deleting a pixel is not possible. We can at best change the value of the pixel and not delete it in images)  \-------------------(3a)
>
> 2. The drop in output probability is proportional to the relevance/importance of pixels, i.e., the output probability will drop to a larger extent for a more important pixel than a less important one.  \-------------------(3b)
>
> Hence, we see that both parts of assumption of perturbation-based saliency metrics (i.e., 3a and 3b) might deviate from point (1) i.e. violating the belief in point (1). We will now show how our proposed approach and metrics helps in quantifying this deviation.
>
> To quantify the deviation of point (3a), we proposed ARPPD, which is the fraction of pixels under investigation with a positive drop in output probability. Thus lower ARPPD would indicate higher deviation of point (3a) from point (1) indicating inconsistency of saliency metrics. ----------------------------------------------------------- (A)
>
> Point 3b talks about the rank list of pixels (Ig) which is different from the rank list of pixels given by saliency maps. 'Ig' is calculated by looking at the amount of drop in output probability by perturbing different pixels and is calculated internally (implicitly) by the saliency metrics. So, in other words the saliency metrics assume Ig as some kind of ground truth and gives a higher score to a saliency map whose pixel ranks is closer to Ig.
>
> However, Ig may or maynot change for different perturbations. This is because although we know that the output probability changes for different perturbations but we dont know if such changes impact Ig. For example, it might happen that for a milder perturbation, the drops in output probability for the pixels a, b and c were 0.01, 0.02 and 0.03 respectively and for a stronger perturbation (like higher sigma is Gaussian blur) the drops in probabilities for a,b and c were 0.3, 0.4 and 0.5. In both the mentioned cases the amount of drops were different but Ig (i.e. the relevance or importance order of a,b and c) remains the same. Similarly, there might be scenarios where the importance order of a,b, and c changes. So, our proposed metric, APRC, quantifies this variation in Ig by looking at a perturbation(Gaussian blur as a case study) and changing its hyper-parameter (sigma). If APRC is 1 then it indicates that Ig remains the same inspite of the changes in the amount of drop in output probability and hence the perturbation based saliency metrics would be consistent. On the other hands if APRC is low then it indicates that the perturbations based saliency metrics would be inconsistent.
>
> That is, APRC quantifies the deviation of point (3b) from point (1). ------------------------------------------(B)
>
> Overall summary: We agree with the reviewer's comment: "predicting exactly how a specific perturbation will change the model's decision is not well understood. But why it changes is well understood." and our paper is based on it. The concluding points i.e. (A) and (B) show that our work is in agreement with the reviewer's comment. Further, Gaussian blur was used as a case study and without the loss of generality, any other perturbation can be used to check its suitability for perturbation based saliency metrics. We hope our study enables the research community to choose correct perturbation for fidelity analysis and it leads to consistent fidelity results.

---

### Official Review · Reviewer_WPYL · 2023-10-31

**Soundness:** 1 poor
**Presentation:** 1 poor
**Contribution:** 1 poor
**Rating:** 3
**Confidence:** 5

**Summary:**

This paper studies the saliency map created by CAM or its variation. Gaussian Blur is applied with different std values to perturb the image saliency. Then two metrics are proposed to measure the fidelity of the saliency map, denoted as Average Ratio of Positive Probability Drops, and Average Pixel Rank Correlation. The use of Gaussian with the proposed metrics is claimed can better measure saliency CAM.

**Strengths:**

This study attempts to analyze an important task in AI, how to understand the behaviour of a model, and how to measure if the understanding is correct.

**Weaknesses:**

The current draft requires some improvement before publishing, it is a little difficult to follow the idea.
The draft attempt to solve the problem in the fidelity measure for CAM, but the presentation did not show this main issue clearly. Some of the paragraphs are not related or unnecessary, please see the questions below. More importantly, the use of Gaussian Blur is not well introduced or explained, it seems an arbitrary solution jumping into the draft. The only reason discussed in Section 3.3 is to preserve the semantic information instead of removing it. However, this is exactly opposite to those previous perturbation used for fidelity measure. If the semantic content is always needed, how can we measure the CAM? or we do we still need to perturb? This draft seems missed the purpose of the perturbation, so the whole story is less solid. Besides the general idea, I highly doubted if Gaussian blur can retain the semantic information when the std is high enough. The two "proposed" metrics are very similar to previous solutions but using the Gaussian perturbation, thus the metrics cannot be considered as new proposals. To sum up, this draft requires more improvement at this moment.

**Questions:**

1. Generally, the abstract and the introduction show the inconsistency in those saliency metric, which is used to measure the fidelity of those XAI methods. However, Fig.1 only shows the tiny difference in the CAM visualization, and this showcase seems not connected to the main story. Is it better to show the behaviour of those metrics when perturbing with different strategies?

2.The definition of the hierarchy in Table is not clear, could be removed.
3. Page 3, third paragraph, "the reason for such unreliability.. lies at model level", this claim is questionable, and I cannot find any explanation for this claim.
4. The same paragraph, "introduce two new metrics,....at model level 1", this is confusing without any explanation, the proposed metrics are also based on the change of confidence scores or pixel rankings, why they are at different levels than the previously proposed? Again, I suggest introducing the idea without relying on the confusing hierarchy could be more clear.
5. The extensive related work is not really necessary, the most related work seems (Tomsett 2020) and those fidelity metrics. However, they are not properly presented regarding the key problem, why different perturbation may lead to different results.
6. Section 2.1, first paragraph is confusing to me. It seems the whole paragraph is repeating how those metrics works, without introducing what the limitation is and why it is a limitation.
7. Section 2.2 Contribution 1: "to justify the unreliability"?? Again, the draft didnot show clearly what the unreliability is, and why does it need to be justified??
8. Section 2.2 Contribution 2: "Unlike...", The proposed methods are very similar to the previously used for fidelity measure, this claim is unclear to me.
9. Section 2.2, Contribution 3: ablation studies on different models and data are not considered as contributions.
10. Section 3.1 "Let R0 be the ranks of pixel", this sentence is wrong and inconsistent with the story after, it is not a list of ranks, it is a list of ranked pixels.
11. Section 3.1, the presented "framework" is not necessary, this is the common assumption behind most of the fidelity measure.
12. Eq(6), why is the rank invariant to the std value applied?
13. what is the meaning of the brackets in Eq 7?
14. For both metric, why are they both approaching 1 ideally? how can we know which saliency map is better? Strong gaussian blur will damage the semantic content, why we are still expecting 1?

Tiny problems:
The use of acronym is inconsistent.

---

> ### Author Response · Authors · 2023-11-21
> **Rebuttal: Official Review of Submission4212 by Reviewer WPYL**
>
> Overall Comment: We thank the reviewer for the helpful comments and suggestions. However, our work seems to be misinterpreted and we respectfully clarify the same. We will use "assumption" to denote "output probability is proportional to the relevance/importance of the pixel" in all subsequent text:
>
> “This paper studies the saliency map created by CAM or its variation.”
>
> Author’s comment: We do not study the saliency maps created by CAM or it variants.
>
> “ Gaussian Blur is applied with different std values to perturb the image saliency. ”
>
> Author’s Comment: We perturb the original image and this is done to validate the assumption. We therefore take Gaussian blur as a case study to validate the assumption.
>
> “Then two metrics are proposed to measure the fidelity of the saliency map, denoted as Average Ratio of Positive Probability Drops, and Average Pixel Rank Correlation.”
>
> Author’s comment: We are proposing ARPPD and APRC metrics to validate the assumption, and not to measure the fidelity of the saliency maps. Our experiments are designed to answer why existing fidelity metrics are were observed as inconsistent.
>
> “The use of Gaussian with the proposed metrics is claimed can better measure saliency CAM.”
>
> Author’s comment: Our study does not propose any fidelity metric for CAM. However, we consider Gaussian blur for illustrating a case study for using our metrics. Our metrics to study the inconsistency can be used with any other perturbation.
>
> ----
>
> Author’s comment for Q1: We focused on understanding the inconsistent behavior in saliency metrics, as already observed by Tomsett et.al, rather than repeating their experiments. Our goal was to understand why such inconsistent behavior was observed. However, we will consider aligning the content to eliminate any confusion and make the message clearer.
>
> Author’s comment for Q2: Without Table 1 it is likely that the reader might confuse our work as another metric to measure fidelity of saliency maps.
>
> Author's comment for Q3: The claim is discussed in detail in the third paragraph in page 3 (i.e., above the mentioned lines). Further, it has been discussed in detail in Section 2.1 and in Section 3 along with the theoretical justification (Sec 3.1) and the proposed metric (Sec 3.2). The low value observed for ARPPD and APRC in the Results and Discussion (Section 5) provide evidence for the unreliability at the model level.
>
> Author's comment for Q4: Saliency metrics use the change in output probability for quantifying the fidelity of the saliency maps. However, we use the same drop in output probability with different formulation (refer Section for saliency metrics and Section 3.2 for proposed metrics) to validate the assumption and provide a score for it.
>
> Author's comment for Q5: We have provided extensive set of related works for the better understanding of the problem. However, we will make it concise by aligning our arguments based on the problem our paper is addressing.
>
> Author's comment for Q6: We discuss the limitation and why it is a limitation in Section 2.1. However, we will revise this section to make the limitation explicit based on the suggestion of the reviewer.
>
> Author's comment for Q7: We'll replace “justify the unreliability” with “investigate the inconsistency” for clarity.
>
> Author's comment for Q8:: Define what is different to previous metrics (levels and what they measure)
>
> The previous metrics like AOPC, Average Drop(AD%), Increase in Confidence(IC%), Win(W%) etc measure the fidelity of saliency maps. These metrics use the observed change in output probability by removing pixels from an image to calculate the fidelity score (as explained in Section 2.1).
>
> Our proposed metrics too use the observed change in output probability before and after perturbation, but we use the change to determine if the assumption holds true or not and quantify it (discussed Section 3 and Section 3.1). The context of “unlike” is to indicate not the observation, but the reason behind the observation.
>
> Author's comment for Q11: The theoretical framework is necessary as it enables us to evaluate if the assumption holds true or not. While we agree that it is a common assumption, we also mention this explicitly in our paper to illustrate the fact that it does not always hold true. Further, the low values of ARPPD and APRC in Section 5 supports that the assumption does not always hold true.
>
> Author's comment for Q12: Rank invariance is key to avoid fidelity score ambiguities due to different perturbations, addressing the inconsistency observed by Tomsett et.al.
>
> Author's comment for Q13: The brackets in Eqn 7 indicate an Indicator function, which we'll elaborate on in the equation's description.
>
> Author's comment for Q14: Ideal ARPPD and APRC values of 1 suggest perturbation invariance, indicating consistent fidelity analysis, regardless of perturbation type.
>
> Author's comment for Q9, Q10 and Q15: Thanks for the comments. We will address them in the final version.

---

> > ### Comment · Reviewer_WPYL · 2023-11-21
> > **response**
> >
> > Thanks the author for answering those questions, however, the answer cannot make the draft clear.
> > 1. "Author’s comment: We do not study the saliency maps created by CAM or it variants."
> > Could you please add one more sentence explaining what the draft is studying? If this is not the case, then the writing is even more confusing, and I could not figure out what Fig1 is for. Is it the assumption that pixel-wise perturbation on CAM maps is not reliable?
> >
> > 2."Author’s Comment: We perturb the original image and this is done to validate the assumption. We therefore take Gaussian blur as a case study to validate the assumption."
> > This answer seems a statement not an explanation. could you please explain what problem this draft is solving? Gaussian Blur to validate what assumption? Pixel-wise measure is not reliable? Why Gaussian can measure validate this and how it is different from the pixel-wise? If the saliency map is pixel-wise, why we have to apply gaussian? The Guassian is also applied pixel-wise, why the std region helps or showcases the problem of the previous perturbation?
> >
> > 3. "We are proposing ARPPD and APRC metrics to validate the assumption, and not to measure the fidelity of the saliency maps. Our experiments are designed to answer why existing fidelity metrics are were observed as inconsistent."
> > I didn't see how the proposed metrics can evaluate the inconsistency. Page 6 3.2, "critical questions 1 and 2", you can use gaussian to show the output is different than pixel-wise attack without explaining why? One can use any perturbation to show a different result but it cannot show why pixel-wise is a problem.
> >
> > 4. "Our study does not propose any fidelity metric for CAM. However, we consider Gaussian blur for illustrating a case study for using our metrics. Our metrics to study the inconsistency can be used with any other perturbation."
> > Then the abstract and introduction can be re-written for a better readability. Maybe it is better to show why pixel-wise attack is a problem then why your metric can showcase the problem. Simply showing "difference" is not meaningful.
> >
> > 6. "Author’s comment for Q1": Again, showing a difference is not informative, showing why pixel-wise perturbation is a problem could be more clear.
> > 7. "Author’s comment for Q2": Maybe this is just to me, it is my first time to see this categorization and it is not clear to me why they are grouped like that. This is not a main problem.
> > 8. "Author’s comment for Q3": This is still questionable to me, why the rank of that pixel has to be invariant to any perturbation? If I apply a Guassian to the top rank pixel, the spatial neighbourhood will also be affected, which makes the rank changes significantly. I cannot see why the assumption is true.
> > 9. "Author’s comment for Q4:"  Isnt "drop in output probability" considered a change in the output probability?? The "proposed" metric is measuring the same property.
> > 10 "Author’s comment for Q8:" Again, I doubt if the assumption holds true, it is only possible to theoretically discuss why the rank change is problem, and why your metric can show this. To me, they are both measuring the drop in the output space based on perturbation on pixels. Or let me raise this question, if I apply Gaussian perturnbation with the old metrics, and the rank changes, can I say the old rank is enough to showcase the "problem"(questionable if it is a problem within the scope of this draft)?
> > 11. "Author's comment for Q14": Ideal ARPPD and APRC values of 1 suggest perturbation invariance, I highly doubt if Gaussian blur can give you invariance in this case, it could make the explanation even more confusing considering other pixels in the neighbourhood into account.

---

> ### Author Response · Authors · 2023-11-22
> **Response to Reviewers questions**
>
> We thank the reviewer for reviewing our comments and putting the next set of questions. We will first humbly describe an example and then answer the questions.
>
> Suppose we want to evaluate the fidelity of saliency maps generated by two CAM methods (GradCAM and GradCAM++) for an image I and model M.
>
> Let’s assume that there are only 6 pixels(for simplicity) in the image which are a,b,c,d,e,f. Based on GradCAM's saliency map, the importance of the pixels(I1) is: a,c,b,d,e,f and based on GradCAM++, the importance ranks of the pixels(I2) is: a,d,f,e,c,b. The importance order is most important first. The task of any saliency metric is to idenity which one of I1 and I2 has higher fidelity.
>
> Suppose the saliency metric uses perturbation technique p1. Based on the drop in probability, the saliency metric calculates a score like AOPC or insertion AUC or deletion AUC etc for I1 and I2. Based on this score, the saliency metric would score either the saliency map from GradGAM (i.e. I1) or the one from GradCAM++(I2) to have higher fidelity. However, the saliency metrics have an implicit assumption that the drop in output probability is proportional to the importance of the pixel (section 2.1).
>
> To explain it further, let Ig be the importance rank of pixels (lg: a,c,d,b,e,f for p1) obtained by perturbing pixels. Ig is different from I1 and I2 as it is calculated internally (or implicitly) by the saliency metrics. Since, l1 is more similar to lg (i.e. higher AOPC or insertion-AUC score etc) as compared to I2 so, I1 would be rated as having higher fidelity. However, for perturbation(p2), lg' might be (a,d, f,e, b,c) and so, for p2, I2 would be rated as having higher fidelity(discussed in section 2.1). Thats is to say, if for different perturbations lg changes, then for some perturbation it would rate I1 as having more fidelity and for some other perturbation it might rate I2 to have more fidelity (discussed in section 2.1). But this inconsistency would not be there if for all perturbations, there is no variation in Ig. In our paper, we show that the inconsistency observed by Tomsett et al 2020 was because of this change in Ig for different perturbations (discussed in 3.1 eqn 6). And so to evaluate the inconsistency we can bypass the analysis of the the saliency maps(i.e. I1 and I2) generated by the CAM methods and directly use ARPPD and APRC(which are derived from section 3.1 eqn 6). This is one positive aspect of our approach i.e. we can evaluate the inconsistency in saliency metrics without analyzing the saliency maps from different CAMs. Although, we use change in output probability (like saliency metrics) but we use it to calculate variation in Ig for different perturbations.
>
> Comment for Q1: We are not studying the saliency maps created by CAM and its variants but we are studying the reason behind the inconsistency of perturbation based saliency metrics to evaluate the fidelity of saliency maps (kindly refer above example). This inconsistency (as observed by Tomsett et al. 2020) was apparent in how different CAM methods were scored by existing saliency metrics, with the scores varying based on the perturbations applied. Consequently, the fidelity of different CAM methods was rated differently under different perturbations.
>
> Comment for Q2: Tomsett et al. 2020 used perturbations like making the pixels 0 and replacing with random valueswhich are quite different. Therefore, we used Gaussian Blur with different sigmas so that the perturbations we use, were similar to each other than those used by Tomsett et al. 2020. If the ranks of the pixels ‘Ig’(calculated by looking at the drop in output probability) vary for small changes in sigma of Gaussian blur then it indicates that the ranks would vary even more with entirely different perturbation technique (like assigning 0 or random value).
>
> Comment for Q3: As observed by the reviewer "One can use any perturbation to show a different result" i.e. based on different perturbations, Ig would be different and as such saliency metrics would rate different CAM method to have higher fidelity (explained in the above example)
>
> Comment for Q4: We will make the abstract and introduction more readable as per the suggestion of the reviewer.
>
> Comment for Q5 and Q7: Kindly refer to the example provided above.
>
> Comment for Q8: The example above discussed different uses of drop in output probability.Gaussian blur is used as a case study but the theoretical justification in section 3.1 and the proposed metrics 3.2 can be used to evaluate if Ig changes, for other types of perturbations too.
>
> Comment for Q9: As observed by the reviewer using stronger perturbations (like Gaussian blur with high sigma) would alter the output probability significantly. Hence, this makes it imperative to choose the perturbation technique carefully and accept the results of fidelity analysis as dependent on perturbation technique used.

---

### Official Review · Reviewer_k88p · 2023-10-31

**Soundness:** 3 good
**Presentation:** 3 good
**Contribution:** 3 good
**Rating:** 8
**Confidence:** 2

**Summary:**

This paper indicates that the existing saliency metrics based on pixel importance are unreliable when subjected to perturbations. To quantify the inconsistencies in saliency metrics, the Average Ratio of Positive Probability Drops(ARPPD) and the Average Pixel Rank Correlation(APRC) are introduced to measure the unreliability at model level.

**Strengths:**

1.	The authors investigated the reason of statistical inconsistencies in the existing saliency metrics which is omitted in previous studies.

2.	Extensive experiments demonstrated the applicability of the proposed metrics.

**Weaknesses:**

1.	The captions in Figure 2 are illegible, and Figure 2 needs more detail legends.

2.	I wonder if the perturbation is similar to some kinds of Data Augmentation, the results may be impacted.

**Questions:**

See weaknesses

---

> ### Author Response · Authors · 2023-11-20
> **Rebuttal: Official Review of Submission4212 by Reviewer k88p**
>
> Overall Comment: We thank the reviewer for the helpful comments and suggestions.
>
>
> Q1: "The captions in Figure 2 are illegible, and Figure 2 needs more detail legends."
>
> Author's comment: Thanks for the feedback. We will take address the points regarding Figure 2 in the final version.
>
>
> Q2: "I wonder if the perturbation is similar to some kinds of Data Augmentation, the results may be impacted."
>
> Author's comment: We acknowledge that different data augmentation, representing different perturbations, can be interesting for future works. In the scope of this work, we focused on the theoretical framework and the proposed metrics which can be easily extended for other perturbations.

---

### Official Review · Reviewer_1hvD · 2023-11-01

**Soundness:** 2 fair
**Presentation:** 1 poor
**Contribution:** 2 fair
**Rating:** 3
**Confidence:** 1

**Summary:**

This paper discusses the problem of disturbance in saliency metric, and holds that pixel importance become unreliable for measuring the fidelity of saliency maps. The authors proposed two metrics to quantify the inconsistencies using prediction probability change and variation of pixel ranks and make two related experimental analysis.

**Strengths:**

1. The problems discussed are of research significance and worthy of attention.
2. Two metric are proposed to analyze the influence of pixel sorting disturbance.

**Weaknesses:**

1. Motivation is not sufficient. It is far-fetched to introduce the necessity of metric research of saliency map from XAI. Especially in Figure 1 and Table 1, the conclusions drawn have limited relevance to the background of the topic introduced in this paper.
2. The argument is not rigorous enough. The article mentioned many times that we should investigate the reasons behind the inconsistency, but we didn't see the relevant analytical statements and experimental analysis in the article, and only gave Figure 1, but Figure 1 can only show that there is no difference between models and it is not directly related to the disturbance emphasized in this paper.
3. The experiment is not reasonable enough. In this paper, a total of 26,267 photos before and after the disturbance are directly tested together, which does not reasonably distinguish the experimental comparison before and after the disturbance, and can not clearly see the inconsistency of metirc after the disturbance.
4. The conclusion is incomplete. From the analysis of arppd in table2, the conclusion of "much more compliant" cannot be drawn by hovering between 0.6 and 0.8. In addition, the final experimental conclusion has no exact correlation with the title of this paper, and the reasons behind it are not given.

**Questions:**

See the weakness listed above

---

> ### Author Response · Authors · 2023-11-20
> **Rebuttal: Official Review of Submission4212 by Reviewer 1hvD**
>
> Overall Comment: We thank the reviewer for the helpful comments and suggestions.
>
> Q1: "Motivation is not sufficient. It is far-fetched to introduce the necessity of metric research of saliency map from XAI. Especially in Figure 1 and Table 1, the conclusions drawn have limited relevance to the background of the topic introduced in this paper."
>
> Author's comment:
> Figure 1 explains the need for saliency metrics as different saliency methods highlight different regions for the same model and the same image to be explained. However, these saliency metrics are not thoroughly analyzed for their reliability   as mentioned by Tomsett et al., 2020. Hence, we take this as the background of our work.
> Table 1 illustrates the 4 levels and helps in explaining why the inconsistencies are not just the property of the saliency metrics but a property of the model itself which is not studied so far . We prove that the inconsistency is a model property by theoretical justifications [Section 3] and the low value of ARPPD and APRC in Results [Section 5]
>
> Q2: "The argument is not rigorous enough. The article mentioned many times that we should investigate the reasons behind the inconsistency, but we didn't see the relevant analytical statements and experimental analysis in the article, and only gave Figure 1, but Figure 1 can only show that there is no difference between models and it is not directly related to the disturbance emphasized in this paper."
>
> Author's comment: Our main argument is to question the validity of the assumption that “Output probability drop is proportionate to the importance of a pixel”. This assumption is central to all metrics mentioned in Section 2 but there has been no study to verify the same . Section 2.1 and in section 3 we propose our framework with theoretical justification and noting the deficiency, we propose two metrics  validate our hypothesis . As a case study, we present experimental analysis with Gaussian Blur to disprove that the mentioned assumption does not hold true for all cases. This explains why Tomsett et al., 2020 observed the inconsistencies.
>
> Q3: "The experiment is not reasonable enough. In this paper, a total of 26,267 photos before and after the disturbance are directly tested together, which does not reasonably distinguish the experimental comparison before and after the disturbance, and can not clearly see the inconsistency of metirc after the disturbance."
>
> Author's comment: The results for each dataset and model explained in table 2 and figure 2 for ARPPD and APRC has been clearly mentioned. As proved in section 3, the metrics ARPPD and APRC capture the impact of applying different perturbations (as a case study we analyzed Gaussian Blur with three sigma values). As per the proof of Sec 3 and Sec 3.1, the ideal value of these metrics should be 1 but as seen in Table 2 and Figure 3, the observed values are far below it.  The observation proves the very assumption (that the drop in output probability is proportional to the importance of the pixel) does not hold true necessitating the careful application of saliency metrics for fidelity analysis.
>
> Q4: "The conclusion is incomplete. From the analysis of arppd in table2, the conclusion of "much more compliant" cannot be drawn by hovering between 0.6 and 0.8. In addition, the final experimental conclusion has no exact correlation with the title of this paper, and the reasons behind it are not given."
>
> Author's comment: The title says: “Why Sanity Check for Saliency Metrics Fails” and in Section 5, we answer this question as: “As seen from Table 2, APRC values are low at each class level and also over the whole dataset. This indicates that there is high variance in the ranks of the pixels for the same model and the same image for different types of perturbations. This would mean that the same pixel would be given different ranks for different perturbations. Thus, the idea of verifying pixel importance ranks of saliency maps leveraging perturbations would be unreliable due to this high variance in pixel ranks. Consequently, all metrics like AOPC and faithfulness etc. that implicitly rely on the invariance of pixel ranks would be rendered ineffective in verifying the fidelity of saliency maps.”.
>
> Further, Eqn 4 for ARPPD measures what fraction of pixels adhered to the assumption that the output probability drops on removing a relevant pixel. An ARPPD score of 0.8 would mean that 80% of the analyzed pixels had a drop in output probability but for 20% there was an increase in the output probability (contrary to our assumption of drop). Hence a higher score is better.  As explained in Section 3, a higher score indicates consistency and a lower score indicates inconsistency in fidelity analysis using saliency metrics.

---

### Meta-Review · Area_Chair_DMr1 · 2023-12-05

**Metareview:**

This work aims to study the consistency of different saliency metrics to perturbations, focusing on perturbations caused by Gaussian blurring. All reviewers agree that the work attempts to analyse an important task. However there were consistent concerns, including from the highest confidence reviewers:

(i) the clarity and organisation of the manuscript needs significant improvement
(ii) the review of related work, formulation of the problem being solved, and proposed analysis is unclear and/or unsound even after discussion with the authors
(iii) the authors claim that their approach applies to any perturbation, without a convincing justification of why.

All of these concerns persisted after the author-reviewer discussion period. The majority recommendation was to not accept the paper in its current form. The AC shares the above concerns and sees no reason to override the majority opinion.

**Justification For Why Not Higher Score:**

See above

**Justification For Why Not Lower Score:**

n/a

---

### Decision · Program_Chairs · 2024-01-16

Reject